# Clinical Manifestation, Transmission, Pathogenesis, and Diagnosis of Monkeypox Virus: A Comprehensive Review

**DOI:** 10.3390/life13020522

**Published:** 2023-02-14

**Authors:** Faheem Anwar, Fatima Haider, Sarmir Khan, Ibrar Ahmad, Naveed Ahmed, Muhammad Imran, Summya Rashid, Zhi-Guang Ren, Saadullah Khattak, Xin-Ying Ji

**Affiliations:** 1Department of Biotechnology and Genetic Engineering, Hazara University, Mansehra 21300, Pakistan; 2Henan International Joint Laboratory for Nuclear Protein Regulation, School of Basic Medical Sciences, Henan University, Kaifeng 475004, China; 3Department of Bioinformatics and Biosciences, Capital University of Science and Technology, Islamabad 44000, Pakistan; 4Center of Reproductive Medicine, Henan Key Laboratory of Reproduction and Genetics, The First Affiliated Hospital of Zhengzhou University, Zhengzhou 450000, China; 5Department of Microbiology and Parasitology, School of Medical Sciences, Universiti Sains Malaysia, Kubang Kerian 16150, Malaysia; 6Institute of Microbiology & Molecular Genetics, University of Punjab, Lahore Campus, Lahore 54000, Pakistan; 7Department of Pharmacology & Toxicology, College of Pharmacy, Prince Sattam Bin Abdulaziz University, P.O. Box 173, Al-Kharj 11942, Saudi Arabia; 8Institutes of Traditional Chinese Medicine, Henan University, Kaifeng 475004, China

**Keywords:** monkeypox virus, emerging viruses, vaccination, pathogenicity, outbreak

## Abstract

Monkeypox virus is a double-stranded DNA virus species that causes disease in humans and mammals. It is a zoonotic virus belongs the genus Orthopoxviral, the family of Poxviridae, associated with the smallpox virus in many aspects. The first human case of monkeypox was reported throughout the Democratic Republic of Congo in 1970. In April 2022, several cases were recorded in widespread regions of Africa, the Northern and western hemispheres. The current review spotlights taxonomic classification, clinical presentations during infection, and the pathogenicity of the monkeypox virus in humans. Furthermore, the current review also highlights different diagnostics used for virus detection.

## 1. Introduction 

The zoonotic virus monkeypox virus (MPXV) was discovered in 1958 after an outbreak of a smallpox-like disease in macaques at a Danish research facility—hence the name monkeypox [1,2]. The first human case of smallpox was identified in 1970 in the Democratic Republic of the Congo, when smallpox surveillance was increasing [3]. During the subsequent ten years, additional monkeypox virus (MPXV) cases were diagnosed, with children accounting for 83% of all cases in the Democratic Republic of the Congo (DRC) and four other Central and West African nations: Nigeria, Sierra Leone, Liberia, and Ivory Coast [4]. Most cases documented in the DRC throughout the 1970s and 1980s had an estimated case fatality rate (CFR) of 11% among persons who had not received a smallpox vaccine. Children under four had the greatest CFR (15%) [5]. Most cases were proven to have occurred in the DRC due to increased surveillance, revealing new MPX-endemic locations in Benin, Cameroon, the Central African Republic, Gabon, Ghana, Sierra Leone, and South Sudan [6]. In 2003, the United States reported the first MPXV outbreak in people outside of Africa, related to an imported exotic pet from Ghana. Seventy-one monkeypox virus cases were reported in individuals exposed to infected prairie dogs, but no deaths were reported [7]. Several nations, including the UK in 2018 to 2022, Israel in 2018, and Singapore in 2019, have recently reported imported MPX cases due to recent travel to sub-Saharan Africa (SSA). The detection of MPX infections in patients in the USA in July 2021 and November 2021 has now been further confirmed by the CDC [8,9,10]. A patient with an unexplained rash who had recently traveled to Nigeria presented at a United Kingdom hospital before 4 May 2022. The polymerase chain reaction (PCR) results on a vesicular swab confirmed the diagnosis of the monkeypox virus [6]. Several more cases were discovered in the following months throughout over 50 countries and six regions, with the largest concentrations occurring in England, Germany, Spain, France, and Portugal [11].

## 2. Features of Monkeypox Epidemics

Monkeypox virus has evolved into two separate clades: West Africa and Congo Basin [12]. Different epidemiological and clinical characteristics distinguish the diseases caused by the two MPXV clades. The Congo Basin strain has a fatality rate of 10% [13]. West Africa has a fatality rate of approximately 1%, with a higher mortality rate among patients with HIV co-infection [14]. From 1986 to 1992, only 13 incidents were reported of monkeypox, and none were recorded from 1993 to 1995 [15]. In 1997, a total of 88 individuals were confirmed to have monkeypox (MPX) infection, whereas in 1996, there was a sudden increase in the number of MPXV-infected human cases reported in DRC [15,16]. MPX outbreak occurred in the United States in 2003. This is the first outbreak of MPX outside of Africa, and it has been linked to the importation of marmots from Africa into the United States. A total of 47 people have been detected in five states [17,18]. From September to December of 2005, Sudan reported ten confirmed cases and nine suspected cases of MPXV due to an outbreak of MPX [19]. Between 2006 and 2007, human MPX infection was again detected in DRC. Since the 1980s, MPX transmission has increased by 20, while smallpox vaccination reduces disease risk by 21 [20].

In addition, monkeypox from Central Africa inhibits T-cell receptor-mediated activation, suppressing proinflammatory cytokine production within human cells from earlier infected monkeypox patients [21]. The MPXV inhibitor supplementing the enzymes or gene inhibits enzymes missing in West African strains and has been recognized as an essential immune modulating factor contributing to the increased virulence of Central African strains [22]. Additionally, the central African strain selectively regulates host responses, particularly in host cell death [23]. Infection with central African monkeypox appears to preferentially inhibit the transcription of genes essential in host immunity, according to transcriptional research [20,24].

## 3. Nomenclature of Monkeypox Virus 

The *Orthopoxvirus* genus of the *Poxviridae* family includes numerous zoonotic infections, particularly monkeypox [25]. To replace monkeypox, the WHO will adopt the new preferred name, “Mpox” [26]. Poxviruses’ primary hosts are lemmings, rabbits, and primates, which can be infrequently passed to humans, allowing the incidence of human-to-human transmission [27]. The *Poxviridae* family is divided into two taxonomic groups: *Entomopoxvirinae* and *Chorodopoxvirinae*. The subfamily classification depends upon whether the virus infects insects, as in *Entemopovirinae*, or vertebrates, as in *Chorodopoxvirinae* [28].

DNA viruses usually increase and express their genomes in the nucleus, making substantial use of cellular proteins; unfortunately, this is not the case for poxviruses [29,30]. The main portion of the genome will contain genes participating the important function, such as virus assembly and transcription, while some are located at termini which participate in virus–host interactions [31]. Poxviruses encode more than 150 genes, and 49 genes are common across all sequence members of the provirus family, while 90 are frequent inside the subfamily of *chorodopoxviruses* [32]. Further, poxvirus genera are divided into eight vertebrates: *Yatapoxvirus*, *Orthopoxvirus*, *Caprioxviru*, *Molluscipoxvirus*, *Suipoxvirus*, *Capripoxvirus*, *Avpoxvirus*, and *Parapoxvirus* (Figure 1). These viruses share the same DNA sequence with the same antigens for reactivity [33].

Poxviruses contain the human smallpox virus, declared eradicated in 1980 by the World Health Organization. Other poxviruses that can infect humans include the cowpox virus, smallpox virus, and Akhmeta virus, as well as viruses from different animal species that can cause a pandemic when combined with the monkeypox virus. The monkeypox virus is the most prevalent of these in humans. Bovine pox is another zoonotic disease with a large population in the animal kingdom, a clear relationship to livestock exposure in humans with variable prevalence and severity and a progression of several weeks. *Paraboxvirus* is a viral genus in the poxvirus family that includes species widespread among sheep, goats, cattle, camels, and the cervix. They are emollients in humans (breast nodules) infected by oral or pharyngeal (papular stomatitis) or skin contact (cellular dermatitis), typically with a moderate granulomatous reaction, causing a variety of clinical manifestations in animals.

Moreover, *Paraboxvirus* results in the formation of large nodular or vascular lesions. The species generates less immunity than *Orthopoxviruses*, increasing the likelihood of reinfection. *Molluscum contagiosum* is a common disease in humans, children, and adults caused by species of the genus *Molluscipoxvirus*, with specific human swellings and erythematous papules mimicking soft tumors, usually with few and variable numbers of immunizations. In the planting area, it is mainly distributed in roots, trunk, and limbs. Immunity is self-limited. The spreading of *Molluscum contagiosum* disease via direct contact with the genital lesions is facilitated by sexual contact and is also spread by Candida.

In contrast to other viruses, smallpox viruses have a much lower mutation ability than RNA viruses, such as influenza viruses. However, these viruses have extensive genetic ability, which explains their ability to attack and evade the virus and the immune system [34,35].

## 4. Transmission

Human monkeypox outbreaks started in the USA in 2003. The first rats were imported from Ghana and sold as exotic pets, which started the trend. These rodents are believed to spread the infection to prairie dogs living in close areas that were also being sold as pets [36,37].

There are multiple routes of transmission for the monkeypox virus (Figure 2), all of which involve direct contact with infected humans or animals. Human infection has been linked to direct animal contact. Still, it is an investigation whether the virus is transmitted through rodent infiltrations or the consumption of bush meat from a common species. Infection with the monkeypox virus in humans is caused by direct contact with animals, especially body fluids such as respiratory fluids and saliva, which may originate from exudates of cutaneous and mucous lesions. Shedding viruses through feces could be an additional route of exposure [38].

In endemic areas of Africa with limited infrastructure and resources, exposure to the feces of infected animals can be a significant risk factor. Many people also choose to sleep outdoors, on the ground, or live close to or visit forests where infected animals are significantly more abundant [39]. In those areas that lack resources and basic needs, such as food, there is no alternative to hunting, which increases the risk of exposure to monkeypox. The transmission rate from animal to animal is higher than human-to-human transmission; respiratory droplets are typically involved, as well as face-to-face and lesion contact among infected individuals [40]. Contact with the contaminated area, such as touching a contaminated object or living, eating, and drinking with the same dishes as an infected person, increases the transmission rate among the population. Due to the possibility of transmission through men who have sex with men (MSM), the monkeypox virus is more prevalent in males than in females during the current epidemic [41,42,43]. In 2022, there is currently a widespread Mpox epidemic affecting several nations in different regions, mainly in the community of MSM, with an appearance that primarily consists of genital lesions [10]. In total, 99% of cases were found in the MSM community in a cohort of 595 confirmed cases of Mpox in Spain in 2022, with the lesions primarily affecting the vaginal, perineal, or perianal regions. Inguinal lymphadenopathy was also shown to be a common characteristic in the research, supporting the idea that sexual transmission was the primary method of infection [42]. As of 6 July 2022, Germany had recorded 1304 confirmed cases, primarily among MSM [44]. Among some verified cases from Nigeria, the findings demonstrate that sexual contact plays a part in the spread of monkeypox [45].

## 5. Morphology and Pathogenesis of Monkeypox

Poxvirus mature structure (Figure 3) has a distinctive dumbbell-shaped nucleoprotein core composed of a double-stranded DNA genome [17]. Monkeypox virions consist of 30 structural and membrane viral proteins, virus-encoded DNA-dependent RNA polymerase, and associated transcription enzymes [46,47]. The virus genome comprises 197,000 bp and contains hairpin termini along with >190 non-overlapping open reading frames (ORFs) [48,49]. The genome’s highly conserved central coding region is guarded by variable ends comprising inverted terminal repeats. At least ninety open reading frames (ORFs) are required for the virion’s replication and morphogenesis. Many non-essential ORFs play a role in the differences in poxvirus host tropism, immunomodulation, and pathogenesis, with many ORFs yet to be functionally categorized [50].

The pathogenicity and pathophysiology of the monkeypox virus begin with virus transmission, which starts with close contact of animal to human or human-to-human. Smallpox and monkeypox infect through the host’s respiratory or oropharyngeal mucosa. The virus enters through the inoculation site and begins replication in the respiratory and oropharyngeal mucosa. Viruses spread to the local lymph nodes in primary viremia. However, these viruses enter lymph nodes and organs via blood circulation in secondary viremia. This process represents the incubation period of 7 to 14 or 21 days [39].

External virion proteins, cellular glycosaminoglycans on the target cell’s surface, and extracellular matrix components are most likely involved in monkeypox virion attachment. Poxviruses enter host cells via a low-pH endosomal pathway or direct fusion with the plasma membrane at neutral pH, releasing the viral core in the cytoplasm. A complex of 12 non-glycosylated viral membrane proteins is required for intracellular mature virions and enveloped extracellular virions to fuse with the cell [51]. Following entry, the virus-encoded multi-subunit DNA-dependent RNA polymerase initiates viral transcription, followed by the translation of early, intermediate, and late proteins on host ribosomes [52]. Poxvirus DNA synthesis occurs in cytoplasmic structures known as “factories”, which gradually transition from compact DNA-containing structures wrapped in Endoplasmic reticulum membranes to crescent-shaped structures where virion assembly takes place. While most mature virions remain within the cell (intracellular mature virions), some are transported via microtubules and enveloped by two Endoplasmic reticulum or Golgi-derived membranes. These enveloped virions can either initiate actin polymerization, which propels the particle on an actin tail toward an adjacent cell, or exit the cell via cytoplasmic membrane fusion and become enveloped extracellular virions (Figure 4) [53].

## 6. Symptoms

In non-human primates, MPVX typically causes a short-lived rash. Plague and 1–4 mm epidermal papules that develop into pustules and crust over are the first clinical signs. A typical lesion has a red, necrotic, and depressed center surrounded by epidermal hyperplasia. These “smallpox pustules” can appear anywhere on the body, but the face and extremities are the most common. Most infected animals recover quickly; however, fatal cases occur, particularly in neonatal monkeys. Following vaccination, MPVX-caused illness was discovered in various rodents, including prairie dogs, dormice, and squirrels. Clinical symptoms in each of these cases (Figure 5) vary, but fever, weight loss, nasal discharge, coughing, respiratory involvement, and mouth ulcers may establish [2,47,54]. MSM have been reported to represent the majority of cases in the current outbreak, often resulting in genital lesions or a vesicular–pustular rash [40,55,56,57,58,59]. The fact that the perineum and vaginal area are commonly affected by the rash suggests that sexual contact may have been the cause of the condition. Monkeypox is often confused with other sexually transmitted infections (STIs), such as granuloma inguinale, molluscum contagiosum, chancre, or herpes simplex infection [60,61].

The clinical features of traditional smallpox with a high mortality rate are classified as typical, altered, variola sine eruption, hemorrhage, and flat type [62]. The hemorrhagic type, which causes widespread bleeding in the skin and mucosal membranes, and the flat type, in which the pustules remain flat, were both frequently fatal. The pathophysiology of these virulent strains of smallpox is poorly studied. Non-human primate models of smallpox and Orthopoxviral infections have been developed utilizing cynomolgus and rhesus macaques [30].

## 7. Diagnostic Tests

Enzyme-linked immunosorbent assay (ELISA) can detect specific IgM and IgG antibodies in the serum of monkeypox-infected patients after 5 and 8 days of infection, respectively, for serological testing. A four-fold increase in serum antibodies can be used to diagnose monkeypox virus infection at both the acute and convalescent stages. Specificity is insufficient due to antigenic cross-reactions between MPXV and other poxviruses. Consequently, this method is inefficient in identifying monkeypox virions and is frequently employed in epidemiologic studies. Similarly, identification by an electronic microscope is not entirely reliable because virions cannot be distinguished morphologically [63].

Real-time polymer chain reaction (RT-PCR) is a genetic method for detecting monkeypox. The conserved regions of extracellular envelope protein gene (B6R), DNA polymerase gene E9L, DNA-dependent RNA polymerase subunit 18 (RPO18) gene, and complement binding protein C3L, F3L, and N3R genes are usually selected as targets for PCR amplification. Furthermore, for MPXV DNA detection, recombinase polymerase amplification (RPA), loop-mediated isothermal amplification (LAMP), and restriction length fragment polymorphism (RFLP) have also been developed for detection purposes [64].

## 8. Immune Response 

The monkeypox virus (MPXV) produces many viral proteins that help it to hide from the body’s immune system (Figure 6). These can interfere with the signaling cascade of pathogen recognition receptors and stop key transcription factors, such as interferon regulatory factor 3 (IRF3) and nuclear factor kappa-B (NF-κB), from turning on inflammatory genes. MPXV can also mess with interferon signaling by blocking interferon-alpha/beta receptors (IFNα/β) from binding or generating IFNα/β and by blocking pathways that are controlled by protein kinase R (PKR). Monkeypox virus also makes proteins that can attack key molecules that cause inflammation, such as Tumor necrosis factor, interferons, interleukin-1 (IL-1), interleukin-18 (IL-18), and interleukin-6 (IL-6). MPXV can also stop apoptosis in infected cells by making many viral proteins that target the apoptotic pathways. The MPXV Zaire strain from Central Africa also makes D14, which prevents the complement cascade from initiating. However, the West African MPXV strain does not make this viral protein. Lastly, MPXV can slow down the activity of natural killer cells and T cells by interfering with the process of their activation [65].

Monkeypox infection results in the accumulation of less dsRNA than vaccinia virus and inhibits the phosphorylation of the pattern recognition immune receptor protein kinase R (PKR) and Eukaryotic Initiation Factor 2 alpha (eIF2), thereby inhibiting the activation of antiviral responses [66]. It has been reported that a homolog of D7L inhibits the proinflammatory cytokine Interleukin 18 (IL-18), which is essential for the control of monkeypox viremia in mice [67]. Zinc-finger antiviral protein (ZAP) selectively targets CpG dinucleotides in RNAs and exerts selective pressure against CpGs in viral genomes [68,69,70]. However, neither the MPXV genome nor mRNAs are suppressed in CpGs. It has been demonstrated that the C16 protein of the vaccinia virus binds ZAP and inhibits its antiviral activity [71], and its homolog in MPXV may play a similar role. Complement control protein (CCP), which prevents the initiation of the complement activation pathway, is another example of an MPXV immunomodulatory [72]. MPXV also inhibits the T cell receptor-mediated activation of antiviral CD^4+^ and CD^8+^ T cells, thereby interfering with adaptive immune responses [73].

## 9. Prevention, Treatment, and Therapeutics 

According to data, the vaccines used for smallpox are the most effective against monkeypox and reduce the severity of clinical symptoms [50]. There are three types of smallpox vaccines in the US strategic stockpile, including ACAM2000 and JYNNEOSTM, licensed for smallpox. The smallpox vaccine produced by Aventis Pasteur could be utilized in an experimental new treatment for smallpox.

JYNNEOSTM, made from modified vaccinia Ankara Bavarian Nordic strain, is a non-replicating and live attenuated Orthopoxviral and was approved by the FDA in September 2019; it is currently used to treat monkeypox and smallpox [51]. Data indicate that vaccination with the vaccinia virus for smallpox is 85 percent effective against monkeypox [51,52]. Europe has approved the smallpox vaccine IMVANEX, although the United Kingdom has been using it off-label to treat cases of monkeypox [53].

ACAM2000^®^ also has a live vaccinia virus. The FDA approved it in August 2017 to replace the Orthopoxviral vaccine Dryvax, which was withdrawn by the manufacturer [54]. ACAM2000 is used for active immunizations against smallpox disease in individuals with a higher probability of infection. The CDC has recommended an urgent approach with an IND procedure, which enables the use of ACAM2000^®^ for non-variola Orthopoxviral infections for the duration of the pandemic [63].

Neutralizing antibodies appear to be an essential component of the immunological mechanisms underlying cross-protection through vaccination against the vaccinia virus. Vaccinating monkeys with a human smallpox vaccine provides protection against monkeypox, which is consistent with the potential of the smallpox vaccine to provide cross-protection for humans against monkeypox [17,18]. Since smallpox vaccines were discontinued in 1978, resistance to various Orthopoxviruses has decreased, particularly in younger individuals lacking vaccinia-induced immunity, and the number of untreated, affected individuals has increased globally. Indeed, there has been an increase in the incidence and geographic spread of human monkeypox in recent years. Percutaneous vaccinia virus infection induces a broad and heterogeneous serum immune response against a broad spectrum of antigenic vaccinia virus features [64]. The antiviral inhibitory activity of serum from immunized individuals is composed of antibodies with various characteristics [65].

### 9.1. Vaccine

The FDA approved Cidofovir for the first time in 1996 to prevent cytomegalovirus eye diseases in patients with acquired immunodeficiency syndrome. Cidofovir has broad antiviral efficacy against many viruses, particularly herpes and adenovirus. Regarding its use in *Orthopoxvirus* infection, cidofovir was used to treat resistant atopic dermatitis in a 28-month-old boy who had developed severe eczema after contact with his father. The boy survived with no long-term consequences [66].

Bricidofovir was approved by the FDA for the first time in June 2021 to treat smallpox injections [67]. It has previously been used in patients with *Cytomegalovirus, Orthopoxvirus*, and adenovirus infections [68,69]. Brincidofovir was used as a combination therapy in a patient who had received a smallpox vaccine and later developed leukemia [70]. Following chemotherapy, the patient was inoculated gradually and treated with various drugs, including six parts pransidovir [71]. In May 2022, the diagnosis and treatment of seven people infected with the monkeypox virus in the United Kingdom were highlighted. In this retrospective study, three patients were given bransidovir, and all three had elevated liver enzymes, a common side effect of bransidovir use that eventually led to treatment discontinuation [71].

The FDA recognized Tecovirimat in 2018 to cure the smallpox virus [72]. The European Medicines Agency approved it for the treatment of cowpox in January 2022. It has been used to treat cowpox-related eye infections in several cases [73,74]. Infection with vaccinia as part of a multidrug regimen. A laboratory worker exposed to the virus was also given the Tecovirimat vaccine [75]. Table 1 shows additional instances where tecovirimat has been used previously. In the case of MPXV infections, tecovirimat was used to treat a patient who had a travel-related case of monkeypox in the United States in 2021. In July 2021, an extended access protocol for the Central African Republic was announced, along with a proposal for 500 tecovirimat curricula to treat monkeypox [76,77].

### 9.2. Vaccinia Immune Globulin Intravenous

The FDA approved vaccinia-resistant globulin intravenous in 2005 to treat the immunization difficulties caused by vaccinia infection [78]. The vaccine insusceptible globulin was given intramuscularly [79]. The VIGIV version was used intravenously in previously reported cases with FDA approval. Many of the patients in Table 1 received VIGIV in addition to antivirals for treating OPXV infections. Vaccinia immune globulin was also given intravenously to a patient with inflammatory bowel disease who had become infected after exposure to a vaccinia–rabies glycoprotein recombinant virus used in animal bait to help to reduce rabies spread in the animal population. Vaccinia immune globulin was also administered intravenously to a patient with inflammatory bowel disease who became infected after exposure to the rabies virus vaccine used in infected animal bait to reduce rabies transmission in animal populations. VIG is also used to treat two patients who developed symptoms of vaccine infection due to a sexual encounter between a smallpox vaccine recipient and one of the case patients [80,81].

## 10. Conclusions

Following the discovery of isolated cases in the western hemisphere, the monkeypox virus, previously restricted to African regions, poses a threat to humans worldwide. Because human-to-human transmission is common via respiratory globules and direct contact with an infected person’s mucocutaneous sores, social isolation and contact tracking are essential. MPX has been confirmed in adults in their twenties. This is due to smallpox vaccine pass-through immunity, which is lost in the elderly and the young population (aged < 45 years) not vaccinated in numerous countries, especially in Europe. This virus develops and replicates in the cytoplasm before entering the main viremia and infecting surrounding lymph nodes. Monkeypox infection can result in encephalitis, dehydration, respiratory distress, and bronchopneumonia. Corneal scarring is the first and most concerning issue, which can lead to vision loss. It is critical to provide the necessary maintenance treatment to reduce the risk of these diseases as much as possible. Supportive therapy, such as moist occlusive bandages, may be used in areas where the rash is concentrated. Organizations are focusing on understanding how these infections emerge intermittently across Europe and the Western hemisphere as new cases of monkeypox are confirmed worldwide. It is critical to investigate any potential treatments as well as understand the full spectrum of each monkeypox symptom as well as the long-term consequences of the virus and symptoms.

## Figures and Tables

**Figure 1 life-13-00522-f001:**
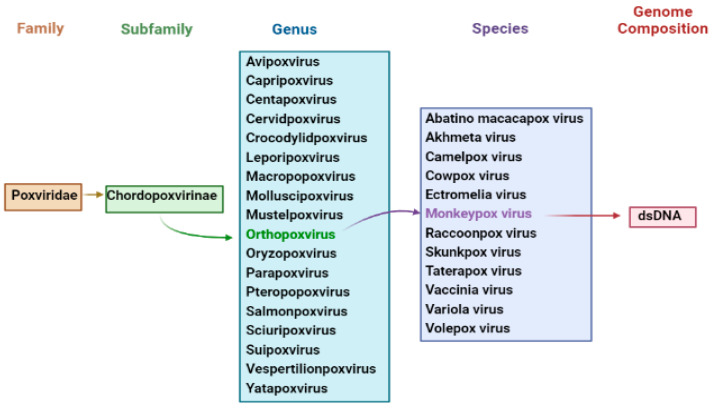
Taxonomy of Monkeypox Virus.

**Figure 2 life-13-00522-f002:**
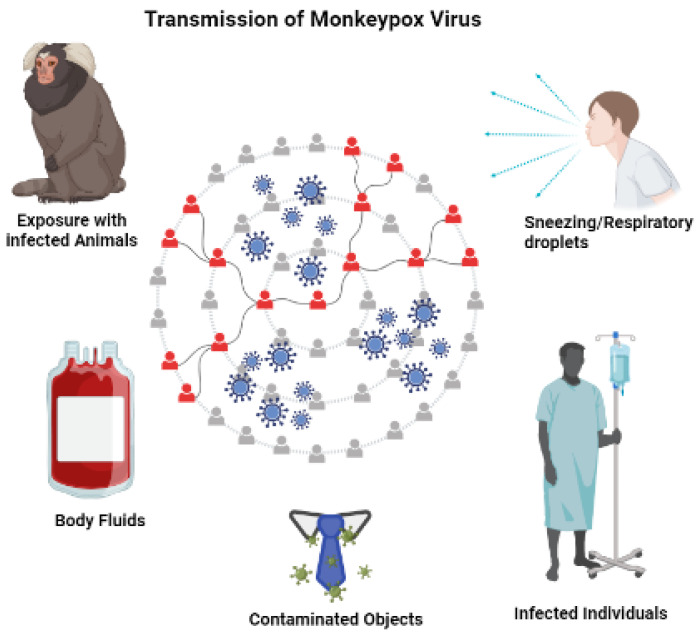
Transmission routes of MPXV.

**Figure 3 life-13-00522-f003:**
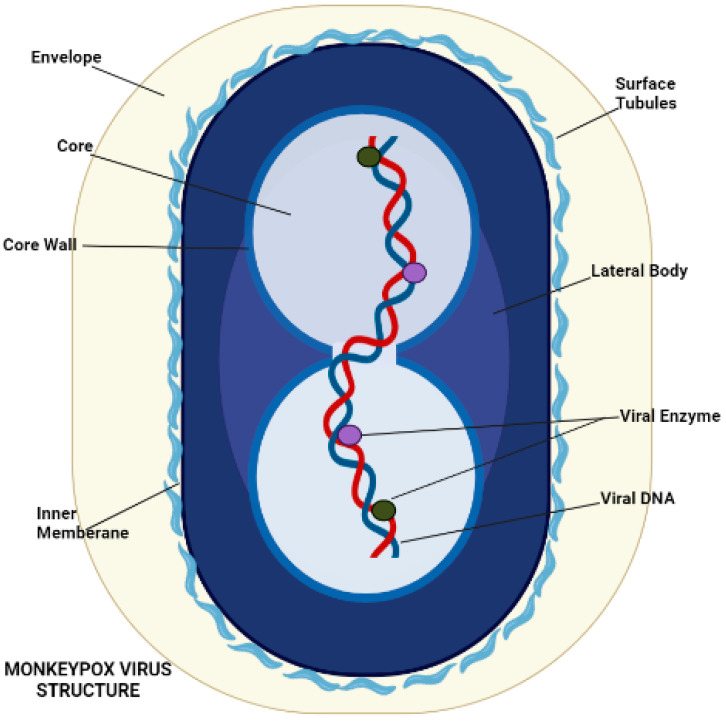
Morphology of Monkeypox Virus.

**Figure 4 life-13-00522-f004:**
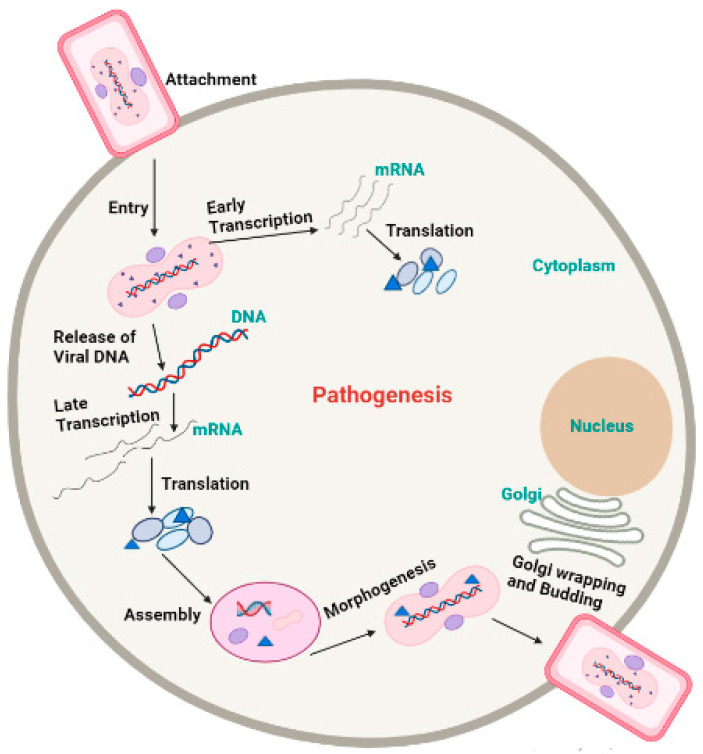
Pathogenesis of Monkeypox virus.

**Figure 5 life-13-00522-f005:**
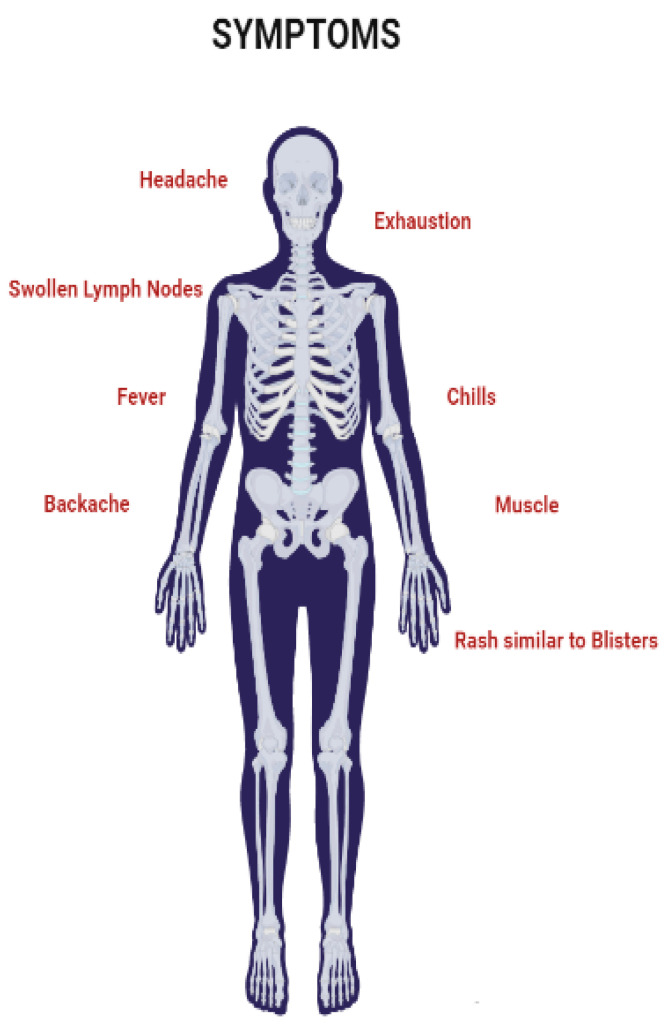
Symptoms of Monkeypox virus.

**Figure 6 life-13-00522-f006:**
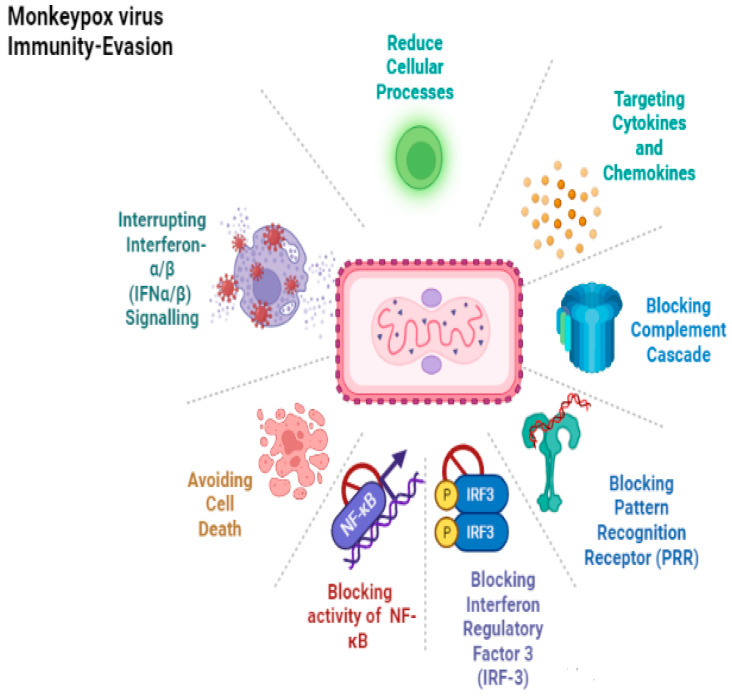
Immunity evasion by Monkeypox Virus.

**Table 1 life-13-00522-t001:** Treatment and Vaccination of Monkeypox Virus.

Treatments	Route	Dosing	Mode of Action	Common Adverse Events	Ref.
Tecovirimat	PO, IV (Approved in May 2022)	Adults: 600 mg twice daily for 14 days; pediatrics (13 kg or more), if 13 kg to less than 25 kg: 200 mg BID for 14 days. If 25 kg to less than 40 kg: 400 mg twice daily for 14 days. If 40 kg or more: 600 mg twice daily for 14 days	Orthopoxviral VP37 envelope wrapping protein inhibitor	Headache, nausea, abdominal pain, vomiting. Infusion-site reactions may occur with IV form	[73,74,75,76,77]
Brincidofovir	PO (tablets, Oral suspension)	Adults weighing ≥ 48 kg: 200 mg once weekly for two doses; adults and pediatric patients weighing ≥10 kg to less than 48 kg: 4 mg/kg of the oral suspension once weekly for two doses; for pediatrics weighing less than 10 kg, the dose is 6 mg/kg of the oral suspension once weekly for two doses	Phosphorylated to the active metabolite, cidofovir diphosphate, which selectively inhibits Orthopoxvirus DNA polymerase-mediated viral DNA synthesis	Diarrhea, nausea, vomiting, and abdominal pain	[67,68,69,70,71]
Cidofovir	IV	5 mg/kg once weekly for two weeks, followed by 5 mg/kg IV once every other week	Undergoes cellular phosphorylation, then selectively inhibits Orthopoxvirus DNA polymerase-mediated viral DNA synthesis	Decreased serum bicarbonate, proteinuria, neutropenia, infection, hypotony of the eye, iritis, uveitis, nephrotoxicity, fever	[66]
Vaccinia immune globulin	IV	6000 U/kg as soon as symptoms appear; may be repeated based on the severity of symptoms and response to treatment; 9000 U/kg may be considered if the patient does not respond to the initial dose	Antibodies obtained from pooled human the plasma of individuals immunized with the smallpox vaccine provide passive immunity	Headache, nausea, rigors, dizziness	[78,79]

## Data Availability

Not applicable.

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
