# Peer review of "Clinical Manifestation, Transmission, Pathogenesis, and Diagnosis of Monkeypox Virus: A Comprehensive Review"

_life, 2023, doi:10.3390/life13020522_

Round 1
Reviewer 1 Report
Dear authors,
Very nice figures. Some concerns below:
1) please the new term of Mpox as proposed in late 2022 by the WHO. it can be introduced in the chapter 3. Nomenclature of Monkeypox Virus
https://www.who.int/news/item/28-11-2022-who-recommends-new-name-for-monkeypox-disease
2) you state "World Health Organization it is unknown whether the monkeypox virus 146 is sexually transmitted or not, but it can be transmitted through close contact [37]." However reference 37 is referring to smallpox in 1988 not the current outbreak. Please clarify.
3) Likewise, please add relevant citation in the literature about genital contaminations and the particular linked to anal lesions in MSM.
One interesting study do hypothesize this fact in heterosexual and should be cited to my opinion. https://pubmed.ncbi.nlm.nih.gov/35969500/
4) In the chapter 6,please cite the different presentation from genitals.
5) In the conclusion you said "This is due to smallpox vaccine pass-through immunity, which is lost in the elderly". Maybe also add the fact that young population (aged <45 years) are not vaccinated at all in numerous countries, especially in Europe.
Author Response
Respected Reviewer, hope that you will be fine.Please find the attachments.Thankyou very much for your kind suggestions and comments.

Reviewer 2 Report
Thank you for sharing the review article on the Monkeypox virus. It is a very well written and comprehensive review. Due to its complexity, please add to the introduction section the overall aim of the review including an outline of the individual topics covered. Please also describe in at least a short methodological section how this review was performed as well as which databases of published literature were used. Here some further comments and suggested edits that could help to improve the article:
L32-33: "where the estimated case fatality rate among unvaccinated individuals of 11%" doesn't seem to be a full sentence; consider revising it.
L34: When were surveillance activities in DRC increased? Consequently, since when was it observed that the majority of cases presumably on a global scale occurred in DRC?
L35-36: During which time frame was/is it observed that the virus is endemic among the countries listed?
L37-38: The content of the sentence isn't fully clear. Was the exotic pet infected with the virus? Did the outbreak occur in the U.S.? How is/was an outbreak defined?
L38-40: Are the 71 cases related to the outbreak reported from the U.S. in 2003 or to another outbreak?
L42: Recently refers to which time frame?
L43: Referring to the Monkeypox cases reported up to L43 of the manuscript, how were they confirmed? Why are the 500 cases named "possible cases"? Were they not confirmed?
L44: Recently or previously? Not clear.
L48: Portuguese or Portugal?
L65-66: Do the mentioned factors relate to Africa only or globally?
L51-66: Consider reporting the epidemiology of the virus in a different manner by using e.g. a table or figure. I would also propose to report cases in a chronological order.
L73-75: The sentence "mutation....throughout infection" isn't fully clear. Consider rephrasing it.
L121: What could be transitional hosts? Please add. L122: Are transitional and intermediate hosts the same?
L123-124: The content of the sentence "imported from ... prairie dogs" isn't clear.
L128: Why is it difficult to find out how the virus can be transmitted? Is it not possible to investigate samples?
L135: Exposure of humans to animal faeces? Please clarify.
L140: Please add figures in terms of higher animal-to-animal than human-to-human transmission.
L144-145: Please add the proportion of Monkeypox cases occurring through sexual practices between men as well as the viral proportion in men versus women.
L162: Also animal-to-animal contact?
Author Response
Respected Editor
We have addressed all the comments and suggestion according to your suggestion and comments.
Thank you very much for your time and considersastion
